# Local and Regional Dynamics of Native Maize Seed Lot Use by Small-Scale Producers and Their Impact on Transgene Presence in Three Mexican States

**DOI:** 10.3390/plants12132514

**Published:** 2023-06-30

**Authors:** Mariana Ayala-Angulo, Edgar J. González, Carolina Ureta, José Luis Chávez-Servia, Emmanuel González-Ortega, Remy Vandame, Alejandro de Ávila-Bloomberg, Geovanni Martínez-Guerra, Said González-Díaz, Rosey Obet Ruíz-González, Prisciliano Diego-Flores, Elena R. Álvarez-Buylla, Alma Piñeyro-Nelson

**Affiliations:** 1Doctorado en Ciencias Agropecuarias, Departamento de Producción Agrícola y Animal, División de Ciencias Biológicas y de la Salud, Universidad Autónoma Metropolitana-Unidad Xochimilco (UAM-X), Ciudad de México 04960, Mexico; mariana.ayala.angulo@gmail.com; 2Departamento de Producción Agrícola y Animal, División de Ciencias Biológicas y de la Salud, Universidad Autónoma Metropolitana-Unidad Xochimilco (UAM-X), Ciudad de México 04960, Mexico; emmanuelgo@iecologia.unam.mx; 3Departamento de Ecología y Recursos Naturales, Facultad de Ciencias, Universidad Nacional Autónoma de México (UNAM), Ciudad de México 04510, Mexico; edgarjgonzalez@ciencias.unam.mx; 4Investigadora por México (Conahcyt)-Instituto de Ciencias de la Atmósfera y Cambio Climático, Universidad Nacional Autónoma de México (UNAM), Ciudad de México 04510, Mexico; carolinaus@atmosfera.unam.mx; 5CIIDIR-Oaxaca, Instituto Politécnico Nacional, Oaxaca 71230, Mexico; jchavezs@ipn.mx (J.L.C.-S.); pdiegoflores@yahoo.com (P.D.-F.); 6Departamento de Agricultura Sociedad y Ambiente, El Colegio de la Frontera Sur (ECOSUR), San Cristóbal de las Casas 29290, Mexico; remy.vandame@gmail.com (R.V.); mcobet.10@gmail.com (R.O.R.-G.); 7Jardín Etnobotánico de Oaxaca, Oaxaca 68000, Mexico; adeavilab@gmail.com (A.d.Á.-B.); geovannimtzguerra@gmail.com (G.M.-G.); 8Maestría en Sociedades Sustentables, División de Ciencias Sociales y Humanidades, Universidad Autónoma Metropolitana-Unidad Xochimilco (UAM-X), Ciudad de México 04960, Mexico; fsaidgonzalez@gmail.com; 9Laboratorio de Genética molecular, Epigenética, Desarrollo y Evolución de Plantas, Departamento de Ecología Funcional, Instituto de Ecología, Universidad Nacional Autónoma de México (UNAM), Ciudad de México 04510, Mexico; 10Centro de Ciencias de la Complejidad (C3), Universidad Nacional Autónoma de México (UNAM), Ciudad de México 04510, Mexico

**Keywords:** conservation, maize, small-scale producers, seed management, native varieties, transgenes

## Abstract

Mexico harbors over 50% of maize’s genetic diversity in the Americas. Native maize varieties are actively managed by small-scale producers within a diverse array of cultivation systems. Seed lot use, exchange and admixture has consequences for the in situ conservation of such varieties. Here we analyze native maize seed management dynamics from 906 small-scale producers surveyed in three Mexican states: Mexico City, Oaxaca and Chiapas. Furthermore, we analyze how their management practices can relate to transgene presence, which was experimentally documented for maize samples associated with the applied surveys. Through a data mining approach, we investigated which practices might be related with a higher probability of transgene presence. The variables found to have a strong spatial association with transgene presence were: for Mexico City, maize producers with larger parcels; for Oaxaca, producer’s age (43–46 years) and the sale of seed; for Chiapas, the use of agricultural machinery and younger producers (37–43 years). Additionally, transgene presence and frequency within the socioeconomic regions of Oaxaca and Chiapas was analyzed. In Oaxaca, higher transgene frequencies occurred in regions where transgene presence had been previously reported. In Chiapas, the border regions with Guatemala as well as a region where reproduction of improved seed takes place, the highest proportion of positive samples were found. A detailed mapping of regional seed markets and seed exchange sites together with deployment of national and local biosecurity measures, could help prevent the further spread of transgenes into native maize varieties, as well as improve conservation efforts.

## 1. Introduction

Maize is a staple in Mexico and occupies the majority of agricultural land sowed to a single crop [1]. The predominance of this crop in the country is not fortuitous; Mexico is the center of origin and diversification of maize, its wild relatives (teosintes) and many other cultivated species [2]. In the case of maize (*Zea mays* L. subsp. *mays*), 59 so-called races and thousands of varieties have been identified, which are strongly associated with Indigenous peoples and campesinos (small-scale producers [3] who use traditional knowledge management practices that dynamically shape maize’s genetic diversity, maintaining alleles that could be necessary to face new selective pressures in response to changing environmental conditions [4]. Thus, several authors posit that the in situ conservation of native varieties in the hands of campesinos, who select and use them according to their needs and culture, giving way to a co-evolutionary biocultural process [5,6] is strategic. In line with this argument, campesino production plays a crucial role in local food security, particularly in rural communities [7,8].

In Mexico, maize is cultivated under complex agricultural dynamics that entail the coexistence of different farming systems under contrasting geo-climatic conditions across the country. Approximately 75–80% of land used for maize cultivation depends on small-scale producers (<5 ha) who tend to use low input, traditional farming methods [9] and predominantly plant native maize varieties, while their production is primarily destined for self-consumption and any surplus is locally sold [8,10,11]. These maize producers commonly save seed from one farming cycle to the next one, and share seeds among themselves, allowing alleles to pass from one generation to another, enabling the evolutionary processes that sustain this crop’s genetic diversity [3,7,8,12]. In the northern part of the country, larger (>10 ha) irrigated maize parcels are predominantly sowed with improved or hybrid seed varieties, while in the south, maize agriculture is largely comprised by rainfed parcels which tend to be sowed, in most cases, with native maize varieties [4,8]. Nevertheless, hybrids are also sowed in rainfed parcels when the agroecological conditions are favorable, as is the case for some parts of the southern state of Chiapas [13]. Furthermore, in the same state or region, contrasting agricultural practices coexist, correlated with Indigenous vs. Mestizo (communities of mixed ancestry) ethnicity, as well as with geography, available infrastructure and plot size.

To attest to the interdependency between this crop and the people that cultivate and conserve it, extant native maize varieties are not homogeneously distributed across the country, being more numerous in the center-south, while their prevalence is highly correlated with regions where various Indigenous peoples live [9,14]. Thus, areas of high landrace richness and diversity have been detected in the mountainous northwest, south central and southern Mexico in the states of Oaxaca, Jalisco, Michoacán and Chiapas [9,15].

The criteria for seed selection used by maize producers are complex and the most important elements relate to seed viability, which leads to the selection of large ears and/or large kernels, as well as traits that define variety type [8]. Commonly, varieties that excel in a particular agroecosystem are exchanged among local producers [16,17]. The dynamics of seed use and exchange among campesinos are a widespread practice that allows for native maize varieties to persist, and it also enables gene flow among different maize varieties, both within a community as well as across different communities [12,18,19,20,21,22].

Maize seed lot use, dispersal, substitution or loss by small-scale producers and campesinos is a central issue for this crop. Many studies have addressed how phenomena such as land reconversion, abandonment, changes in the demographics of rural households and migration, as well as environmental phenomena such as soil degradation, erratic rainfall patterns, desertification and overall climate change, can negatively affect the in situ use and conservation of native maize varieties in Mexico [3,4,8,10,20,22,23,24,25,26]. Furthermore, it has been documented that the number of native maize varieties present in Mexico has decreased over the last several decades due to several factors that converge, among them, the introduction of hybrid varieties, a phenomenon that after 60 years has impacted the genetic diversity of native maize varieties [27]. Improved varieties (including genetically modified maize) are typically bred for high productivity within a relatively narrow range of agronomic settings, and tend to be more sensitive to poor agricultural conditions than more genetically diverse, open-pollinated native maize [17]. Furthermore, complex dynamics of maize seed/grain distribution allow for seed flow to span thousands of kilometers, and seed replacement can alter local allele frequencies rapidly and decisively [11].

In this context, transgenic or genetically modified (GM) maize is another category of professionally ‘improved’ maize whose introduction into the country has generated controversy due to its potential negative impacts on native maize use and conservation, as well as the agricultural and cultural practices associated with it [28]. In 1996, the first large-scale introduction of commercially cultivated transgenic maize took place in the United States. Since then, adoption of GM maize and other GM crops has increased, to now encompass a global planted area of 185.6 million hectares (2020 data) [29].

Mexico is self-sufficient in the production of white maize, which is a staple of the Mexican diet. In contrast, the country is deficient in yellow maize production [30] hence the country imports between 10 and 12 million tons per year of yellow maize grain [30,31], principally from the United States, where approximately 95% of the maize grain produced is GM [32,33]. According to Reyes and collaborators (2022), in the 2010–2020 period, the average apparent national consumption of maize was 22 million tons; for yellow maize alone it was 14 million, spurring the importation of 11.3 million tons of this type of grain to satisfy demand [34]. Currently, there is scant information pertaining to the path followed by imported and domestic grain, but they tend to be admixed, as the Mexican government does not segregate imported transgenic grain [32,35].

The discussion about the potential consequences of unintended transgene flow into Mexican native maize varieties burst into the public arena after the first report documenting transgene presence in maize samples from Oaxaca [36]. Afterwards, several efforts to monitor for the presence of transgenes in native maize varieties took place in Oaxaca [37] and [38] as well as in other places in Mexico [16]. In all the aforementioned efforts, except for one [39], transgenes were detected, albeit in low frequencies.

Few studies have focused on investigating the extent to which seed management practices are in fact correlated with an increase in the probability of transgene presence in native maize seed lots [16,38]. To our knowledge, the investigation by Dyer and collaborators [16] is to date, the only one that has analyzed seed management practices of a representative sample of rural households across Mexico, through the application of a survey and the collection of maize seed samples. In their work, they found that almost 3% of farmers planted maize grain obtained from a nationwide grain distributor (Diconsa), while informal seed networks based on farmer to farmer seed exchange were the main source of seed flow throughout Mexico, particularly in the south, whereas in the north, farmers tended to purchase hybrid seed from seed companies and rarely re-used it [16].

Here, we present a detailed analysis of maize seed management and agricultural practices by small-scale maize producers that cultivate native maize varieties, surveyed in three states: Mexico City, Oaxaca and Chiapas. We hypothesize that some of their agricultural practices could be correlated with the presence of transgenes detected in some of their seed lots [16]; see Table 1.

These three states have contrasting geo-climatic conditions and agricultural dynamics. While in Mexico City maize production is performed under rainfed conditions and encompasses a relatively small area of approximately 3965 ha [40,41], it nevertheless harbors six native maize landraces [41]. In Oaxaca, a state with 35 landraces already identified [42], a majority of maize is sown under traditional farming systems that use locally sourced maize seed, with low inputs and little or no machinery [43,44]. Chiapas harbors 23 native maize landraces [43] and has a large area planted with maize (690,653 ha in 2020) [45]. In this state, market-oriented plantings are sowed mainly with hybrid seeds, with the use of agricultural machinery, encompassing the municipalities of Villaflores, Ocozocoautla, Cintalapa, and irrigated zones such as La Concordia and Chamic, which are part of the Fraylesca region [46,47].

Through a data mining approach, we explored which seed management practices could be associated with a higher likelihood of finding transgenic sequences in native maize samples, contrasting their differences and commonalities, while also analyzing differences among regions within each state (see Figure 1). We discuss our findings in light of local agricultural practices and point to some factors that could stimulate or hinder transgene flow into native maize varieties, putting forth suggestions that could help improve national and local biosafety measures and which could be applied to reduce both transgene flow into native maize varieties in Mexico, as well as improve in situ conservation efforts.

## 2. Materials and Methods

### 2.1. Sampling Strategy, Maize Seed Lot Collection and Producer Surveys

In the context of a commissioned investigation by the Institute of Ecology and Climate Change (INECC, in Spanish) to one of the authors (E.R. Álvarez-Buylla) involving the design of a representative sampling strategy for transgene detection in native maize varieties, we elaborated an algorithm aimed to use publicly available agriculture data from government agencies (INEGI, SIAP, CONABIO) pertaining to the rainfed area sowed with maize in each municipality. A geographically representative sampling scheme was designed, where the number of localities to sample per state were proportional to the total rainfed area sowed with this crop, per state (for more details, see [48]). This algorithm was tested in five states of Mexico. The three states that had a better sampling effort are analyzed here: Mexico City, Oaxaca and Chiapas, where 20, 258 and 151 localities were sampled, respectively. Field work was performed between November 2017 and November 2018. On average, three maize producers were surveyed per locality. Ten maize ears or 500 g of seed were collected per seed lot and donated by each maize producer, while producers were surveyed through a standardized survey, designed to gain insight into agricultural and seed management practices that could hinder or favor transgene flow into their native maize varieties (see Appendix A). The total number of producers interviewed was 115 in Mexico City, 400 in Oaxaca and 391 in Chiapas.

DNA purified from pooled samples for all maize seed lots collected was evaluated for the presence of transgenes through Q-PCR using gene-specific probes and *Taqman*^®^ chemistry. Transgenic marker sequences assessed were the 35S Promoter from the Cauliflower Mosaic Virus (CaMV P35S) and the Nopaline-synthase Terminator from *Agrobacterium tumefaciens* (T-NOS). Results of molecular analyses were previously published [48] and can be reviewed in Appendix A. Detailed laboratory methods can be consulted in the technical report delivered to INECC [48], while a detailed cartographic analysis of transgene distribution and its relation with biophysical and large-scale social factors has been previously published [48]. In this work, we focus on analyzing the cultural practices documented through a survey instrument (see Table 1).

### 2.2. Interviews: Type and Systematization

Our survey and seed collection efforts were focused on small-scale maize producers or campesinos. The focus on this demographic was due to the fact that the original study was conducted to assess transgene presence in native maize varieties, as they represent the living reservoir of maize genetic diversity in Mexico and are the subjects of different agrobiodiversity conservation efforts. Maize producers who cultivated in rainfed parcels were the target group, under the assumption that most native maize varieties are sowed in this agricultural setting, which is supported by different scholars [3,14].

The survey (Appendix A) included 13 questions (aside from personal information pertaining to name, age, telephone number and self-adscription to an Indigenous community), focused on documenting seed management practices, such as estimated age of the seed lot used, introduction of seed external to the community (native, hybrid or other kind), type of agriculture practiced (monoculture, polyculture), level of technification, use of herbicides and synthetic fertilizers, among other queries. In order to avoid a possible duplication of surveyed localities, each locality was georeferenced (see Figure 1). As a means to explore which factors could be associated with the presence of transgenes in each state, as well as to unravel general trends among producers from the three states under analysis, we generated a single database in which the answers provided by maize producers were encoded into discrete categories (see Table 1). Then, we analyzed the relationship between variables and transgene presence (see Table 2). After statewide analyses, we performed a detailed analysis within the sub-regions for the states of Oaxaca and Chiapas, interpreting our data in the light of regional dynamics (see Table 3).

Briefly, all the participating campesinos who speak an Indigenous language were classified in a single category and participants self-considered as Mestizos were classified in another. The self-reported area of their parcels was classified from 1 to 6 ha; varieties of maize were classified as being either native or hybrid. Uses of maize grain were classified as follows: for food, for fodder or for sale. Venues for maize seed acquisition were classified as: from a seed shop, from the same locality and from another locality. For the question related to the use of fertilizers, herbicides and insecticides, answers were classified as yes or no (see Appendix A).

### 2.3. Data Mining Analyses Performed on Maize Producers’ Survey Data

The answers from interviewed producers had been previously analyzed using general descriptive statistics [48]; data are summarized in Table 1. In order to further analyze these data, we used a spatial data mining approach to explore for potential geographic associations among multiple variables and the presence/absence of transgenes, implementing a method based on Bayesian probability [49] that has already been implemented on native maize [9,50]. We used publicly available data layers related with geographical, biological and social variables, as well as variables obtained from our survey related with seed management and agronomic practices. Through this approach, two values are obtained: epsilon (**ε**) and score (**S**). Epsilon values identify the potential spatial relationship between our study object (i.e., transgene presence) and a variable. In the case of ε, values above 2 suggest a strong spatial association between a particular variable and transgene presence, as this threshold corresponds to two standard deviations in a 95% confidence interval [49]. The score is the probability of finding our study object (i.e., transgene presence) given the presence of specific variables’ categories in a geographic space. We did not find high score values (which could have yielded a predictive capacity) with the variables used, thus we decided to focus on epsilon (ε) values (potential spatial relations), analyzing geographic relations between transgene presence and the variables extracted from our surveys.

## 3. Results and Discussion

### 3.1. Commonalities and Differences in Agricultural and Seed Management Practices among Maize Producers

With regard to the demographic profile of participating maize producers, some characteristics across the three states were similar, such as the age of participants, which averaged 54.6 years; Mexico City and Oaxaca having a minimum average age difference (56.6 and 56.2, respectively), with the percentage of surveyed producers above 50 years old being over 50%, while in Chiapas, a producer’s average age was slightly younger, with a mean of 51.2 years old (see Table 1). We found that the percentage of speakers of a native language was substantially above the reported average in two of the three states: in Chiapas, 22.79% of interviewees spoke an Indigenous language, which is slightly above that state’s average (20.6%) [45]. In Oaxaca, the percentage of speakers of an Indigenous language was almost double the state average (55% interviewees vs. 28.2% statewide) [45]. Notably, in Mexico City, it was almost ten times higher (12.1% of producers vs. 1.32%) [45]. This observation is consistent with multiple studies that have documented the prevalence of Indigenous peoples in rural, rather than urban areas [8,51] and its prevalence is highly related to the conservation and distribution of native maize varieties throughout the territory [14].

Regarding seed use among maize producers from the three states, we found that the majority of maize samples were described as native varieties by their owners (89%) [48]; the remainder was composed, in general, by hybrid varieties. Furthermore, the distribution of native vs. hybrid varieties was not homogeneous: in Chiapas, 77.6% of samples were defined as native varieties, a percentage substantially lower than the samples collected from Oaxaca and Mexico City, where over 90% of maize samples comprised native maize (92% and 98%, respectively). Furthermore, the use of native seed was related with the possession and use of seed lots for a longer time span (9 or more years) and having a restricted seed exchange network where producers primarily exchange seeds with family members and/or neighbors from the same locality (see Table 2).

Notably, although in the three states surveyed, local seed exchange is a common cultural practice, producers from Mexico City have a much higher seed exchange rate outside of their locality (15.6%). This practice did not translate into a higher frequency of transgenes in maize seed lots from Mexico City, which accounted for only 2%, in contrast with the other two states, where we detected higher frequencies, 7% in Oaxaca and 13% in Chiapas; see Appendix A for details and [48]. This phenomenon of a higher rate of incorporation of external seed but low transgene frequencies could be due to two possibilities: one, that such external seed is non-transgenic and is similar to the native varieties that producers already have and as such, is not regarded as an external seed lot; the other possibility relates to a previously documented agronomic practice: the purchase of seed from an external source seed that can be native or from an improved variety planting it, but harvesting it early as immature ears to be sold for consumption as a vegetable within the locality or in the urban areas of Mexico City [52]. Thus, these varieties could pollinate local native varieties, but are not saved as seed and/or incorporated into native seed lots and are not considered by producers as ‘their’ seed lots.

With regard to the diversity of native seed lots owned and managed by producers, in several regions of Oaxaca and Chiapas, most of them sowed a single seed lot of maize (farmer’s variety), while in Mexico City, only 36.52% of producers sowed a single variety and cultivated up to four different varieties of maize. This result was somewhat surprising, given the fact that Oaxaca and Chiapas bear the majority of native maize races in Mexico [3], while in Mexico City, only 6 races have been documented [41]. This could be due to the use of diverse, locally adapted maize landraces sown in particular agroecosystems in Oaxaca and Chiapas [41], as well as drought causing the loss of local varieties. We did not perform landrace comparative studies which aimed to identify the varieties and corroborate that diverse landraces were collected.

Most of the producers in Oaxaca and Mexico City cultivate their maize varieties under a poly crop or ‘milpa’ system, while in Chiapas, maize monoculture is prevalent, particularly in regions with the highest maize production (Fraylesca and Soconusco) (see Figure 1 and Table 3). Most of the maize producers from Oaxaca stated that they have preserved their seed lots and cultural practices over generations, which indicates that tradition favors the conservation of native varieties. In Oaxaca, these varieties are kept mainly due to particular culinary uses [8,22].

In the case of Chiapas, which is a state geographically divided into 15 regions, commercially oriented maize plantings are primarily produced in the Fraylesca and Soconusco regions, and the use of hybrid seeds is common [47] and is produced under monocropping (88%). In consequence, our survey indicated that farmers used more agricultural machinery tractors, compared to Mexico City and Oaxaca (see Table 1). Furthermore, in this state, all producers surveyed applied fertilizers and herbicides to their production, a common agronomic practice in monocrop agriculture (see Table 1). It is noteworthy that, while the average parcel size across the three states was 1.29 ha, when analyzing the two most productive municipalities in Chiapas, Villaflores and Villa Corzo in the Fraylesca region, the average parcel size was 5 ha, consistent with previous estimations [53]. However, we may have incurred a sampling bias for this state, as the algorithm developed in this study to generate the sampling scheme was constructed using governmental data. Due to a lack of official data pertaining to maize producers based in the northeastern part of Chiapas within and around the Lacandon rainforest, this region was not sampled, but as some authors note, this part of the state is inhabited predominantly by different Indigenous peoples who do cultivate maize in an estimated area of 3847 ha [54].

### 3.2. Data Mining Analysis

The data mining approach geared towards analyzing particular spatial associations between transgene presence and agricultural practices as implemented by [50] and modified from [49], yielded few state-specific associations. For Mexico City and Chiapas, only one variable had a strong association (ε ≥ 2) in each state (see Table 2). In Mexico City, having parcels of up to 5 ha was associated with transgene presence (ε = 2.58), while in Chiapas, the variable related to seed acquisition, in particular regarding seed lots acquired from people outside of the locality, had a spatial association with transgene presence (see Table 2). In the case of Oaxaca, two variables were positively associated with transgene presence: producer’s age being between 43 and 46 years old (ε = 2.96) and the destiny of harvested grain, namely, when it was produced to be sold (ε = 2.37). In the case of Oaxaca, our first finding could relate to younger producers having more access to new agricultural technologies, such as hybrid varieties, as well as not keeping seed for a long time or that many of them were or are migrants from the USA. These phenomena must be further studied. Other variables that were close to being significant in both Oaxaca and Chiapas, were producer age group and the age of the seed stock used (see Table 2). Although we lacked predictive power due to low score values, the 13 variables extracted from our survey data yielded a higher ɛ value (spatial association) in comparison with the study from Ureta and colleagues [50], in which 61 large-scale social and environmental variables were used.

As a means to strengthen the values of epsilon regarding variables that were close to being significant, it could be desirable to increase the number of localities per state, in order to increase spatial representation. This is evident in our dataset; while the number of producers surveyed in Oaxaca and Chiapas was very similar (400 and 391, respectively); in Oaxaca, 258 localities were sampled, in contrast with 151 in Chiapas [48]. Additionally, in order to gain further insight into seed management practices, some questions of the survey could be refined. For instance, future works could include a question that explicitly asks if grain from Diconsa or other venues that sell external grain for human and animal consumption have been sown and/or incorporated to a producer’s seed lots. This is a relevant question, as grain which moved through the Diconsa distribution system is biologically viable and can germinate and eventually shed pollen [55]. Another venue of external seed that is locally distributed is through different governmental social programs that commonly give seed to small-scale producers; this is particularly common with producers whose land tenancy falls into the ejido or communal property [35,56]. Furthermore, some of the donated seed includes hybrids, which in some cases bear transgenes [55]. Additionally, the color of introduced seed could be systematically recorded, as transgenic maize produced in the USA is mostly yellow, although white transgenic maize is also available and produced in South Africa, from where Mexico has imported grain [28]. Lastly, another phenomenon that was not explicitly addressed in our survey and could favor the introduction of external seed, is the incorporation of seed brought by immigrant workers who could introduce them into their localities of origin [13]. This possibility has been put forth in multiple works [16,36,37,57,58] among others but remains to be further investigated.

### 3.3. Analysis of within State Regional Variation on Transgene Presence

After a general analysis of the statewide dynamics documented through our surveys, we analyzed our data dividing it into the socioeconomic regions of Oaxaca and Chiapas (see Figure 1). In Mexico City, we did not carry out regional analyses because the number of samples and localities was very small. We did not apply a data mining approach to this regional analysis, as the partition of the original dataset hindered a large enough sample.

When analyzing the frequency and distribution of positive samples within particular regions in Oaxaca and Chiapas (see Figure 1), we found five regions in Chiapas (Metropolitana, Valles Zoque, Meseta Comiteca Tojolabal, Sierra Mariscal and Soconusco) and three in Oaxaca (Sierra Norte, Mixteca and Papaloapan) where the percentage of positives per region was higher than the state’s average (see Table 3).

In the state of Chiapas, the improved seed sowed in these regions is planted in parcels that go from 5 and up to 20 ha, and the three regions with the highest use of such seed are: Centro with 19.2%, Fraylesca with 17.6% and Selva with 12.9%. With respect to Fraylesca, a region that accounts for 17% of the statewide production of maize in 10.7% of the agricultural land in Chiapas, the climate is warm and semi-warm with abundant rains in summer and thus amenable to the use of improved seed varieties. In this region, there are three types of land, depending on the slope: plains or shallows that represent 10% of the total surface, terraces and parcels with up to 20 degrees of inclination. The latter are the most common (34% of sowable land) and thus producers prefer the use of hybrids coupled with the use of machinery. Within this region, the municipalities of Villaflores and Villa Corzo stand out with an area cultivated to maize of 26, 113 and 21,744 ha, respectively [59].

It is noteworthy that the Meseta Comiteca Tojolabal, Sierra Mariscal and Soconusco regions border with Guatemala (see Figure 1) and encompass a prime area for agricultural production geared for export, with an ongoing process of land reconversion and copious use of agrochemicals [60]. In this region, the cultivation of hybrids is primarily destined for export to the United States [61,62,63]. We consider that porous border dynamics could favor the smuggling of transgenic maize seed from Guatemala or other Central American countries into various borderline regions in Chiapas, a phenomenon that has been already reported for other states in Mexico, such as Chihuahua in the north [64] and recently, Tabasco in the south (data unpublished).

Moreover, in some regions, producers are benefited by government programs (MasAgro, ASERCA and “Fomento a la Agricultura”) aimed at increasing production, mainly in Villa Corzo and Villaflores in Chiapas [53] (1) where 37.2% of maize production is hybrid maize. For example, in Villaflores, 75% of maize is produced using only hybrids, while the remaining 25% uses both local and hybrid varieties, dynamics that could favor the introduction of transgenes. In other regions, such as Altos and Selva, maize is also very important, but is sowed as part of diversified small-scale agriculture systems fundamental for local food security.

In the case of Oaxaca, the majority of maize growing areas are rainfed with low mechanization rates and with agronomic practices that include conservation tillage, cover crops, organic fertilization and the use of locally adapted native varieties which are used in the preparation of traditional foods. Additionally, the majority of maize producers are over 60 years of age, who preserve a traditional system for maize production [65]. In Oaxaca, there are some governmental programs focused on the distribution of fertilizers as a means to increase production. Nevertheless, the majority of maize production is based on the use of native varieties, destined mainly for self-consumption, and the surplus is sold in local markets [8]. The regions of Oaxaca with the highest maize production are Mixteca, Valles Centrales, Sierra Sur and Istmo. For this state, the largest number of samples positive for transgenes were in the Sierra Norte, Mixteca and Papaloapan regions. Within these territories, there are important local markets, where the commercialization of maize seed is common, such as Ixtlán de Juárez, Villa Alta, San Pedro and San Pablo Ayutla Mixe in Sierra Norte; Tlaxiaco, Huajuapan, Nochixtlán and Santiago Juxtlahuaca close to Mixteca and Tuxtepec; San Luca Ojitlán, as well as Valle Nacional close to Papaloapan, towns where people can acquire seed for the following agricultural cycles (see Figure 1, Table 3). The Sierra Norte region comprises the districts of Villa Alta, Mixe and Ixtlán. The Mixteca region includes the districts of Coixtlahuaca, Huajapan, Juxtlahuaca, Nochixtlán, Silacayoapan, Teposcolula and Tlaxiaco with high commercial interchange with Puebla (a neighboring state) and Mexico City, where the sale of commercial varieties of maize is a common practice. In Oaxaca, an atmospheric phenomenon locally called the “canícula” or intra-summer drought, is often present between the months of July and August. Depending on the severity and duration of the canícula, maize producers may partially or totally lose their harvest. This situation forces them to acquire external seed for the next planting season. Finally, in the Papaloapan region within Tuxtepec and Choapan districts, maize planted is mainly “tepecintle” [66], Tuxpeño race and different hybrids [67]. Given that producers in this region can plant twice a year with these types of maize, the district contributes 31% of the state’s agricultural production [68]. Additionally, in 2017, maize cultivars were affected by a tropical storm, a phenomenon that caused many producers to lose their harvest and were forced to seek out new seed sources [69]. In this region, a high number of hybrids (simple, double and triple crosses) are commercialized and we estimate that every year around 20,000 tons of seed are used for the two crop cycle [13].

As suggested by the authors in [56], an important source of dispersal of transgenes are the seedbanks, and although we obtained seed samples in municipalities where the seed banks are located, these samples resulted negative for transgenes. However, it is a common practice that producers, having no seed to sow, move to the main regional municipalities (where the seedbanks are located) such as Tapachula, Tuxtla Gutiérrez (Chiapas) and Pinotepa Nacional, San Bartolo Coyotepec and Santo Domingo Tehuantepec (Oaxaca), to stock up on seed to be able to sow.

With regards to the type of transgenes detected among the three states surveyed, more positives were found for the T-NOS than for the P35S sequence, a phenomenon consistent with other recent studies [70,71] (Appendix A). While the study by Dyer and colleagues [16] detected a comparable frequency of transgenes in the southeast of Mexico (5.5%), as the one reported here for Oaxaca (see Table 2), the molecular technique used—ELISA (Enzyme-Linked Immunosorbent Assay) tests—is not directly comparable, as it is based on protein expression and our assays were based on DNA. In this study, few samples were positive for specific transgenic events (NK603: 7, TC1507: 6, MON810: 3), in contrast with [70]. This could be explained by the results obtained from the identification of specific transgenic maize events present in domestic food samples collected, which were compared with the presence of transgenes in food samples produced abroad that were also analyzed, being both categories equivalent with regard to the maize-specific transgenes detected. In this work, the contrasting small number of samples that resulted positive for the identification of specific transgenic maize events could be due to the detection of non-commercial events (see, for example, [72]). This should be further addressed through additional molecular methods, either probing for a larger array of maize-specific commercial events and/or performing next generation sequencing efforts focused on unraveling the composition of the transgenic locus or loci involved. Nevertheless, the report from which this work stems is the first effort to conduct a statewide, representative sampling scheme together with a producer survey pertaining to agronomic practices that could favor gene flow through seed exchange [48]. As such, it sets a reference for future comparative studies and is the first work to detect transgenes in native maize from Chiapas, while it corroborates previous studies addressing transgene presence in Mexico City [57] and Oaxaca [17,36,37]. Our findings suggest that an update of public policies regarding GM maize is needed, including periodic sampling of hybrid maize sold for cultivation as well as imported grain, analyzing both matrices for transgene presence. A special focus should be put on labeling imported maize grain, which can be a continuous source of transgenes if sowed. Additionally, region-specific maize seed exchange, purchase and movement should be further analyzed, as our data suggest that regionally important seed producers or vendors can affect what maize is sowed nearby.

Moreover, GM maize presence in native varieties can become a risk in the form of genetic erosion or in the worst case scenario, lead to genetic assimilation (crop alleles replacing wild ones) as well as causing other unintended consequences, such as the evolution of new or invasive species [73].

Lastly, another pressing subject is the risk of local extinction of native maize varieties in different parts of the country, especially in areas where large-scale maize monoculture is prevalent, as well as areas where demographic and cultural changes of campesinos are taking place. In order to contend with this trend, federal and local efforts to stimulate the sowing of locally adapted native maize varieties and thus, their active in situ conservation, need to be implemented. The latter, not only because these varieties have a great capacity to withstand environmental disturbances, but because they represent an important reservoir of alleles that can help adapt maize agriculture to changing environmental conditions such as drought [73]. This could be accomplished through subsidies to native maize producers as well as through commercialization of native maize grain in both local and urban settings in Mexico. Finally, we consider that the agricultural practices used by small-scale producers and campesinos with regard to their native maize varieties represent a biocultural heritage that should be actively stimulated, together with a participatory approach aimed at the implementation of locally tailored and culturally pertinent biosecurity measures geared towards diminishing the introduction of genetically modified maize into local native varieties.

## Figures and Tables

**Figure 1 plants-12-02514-f001:**
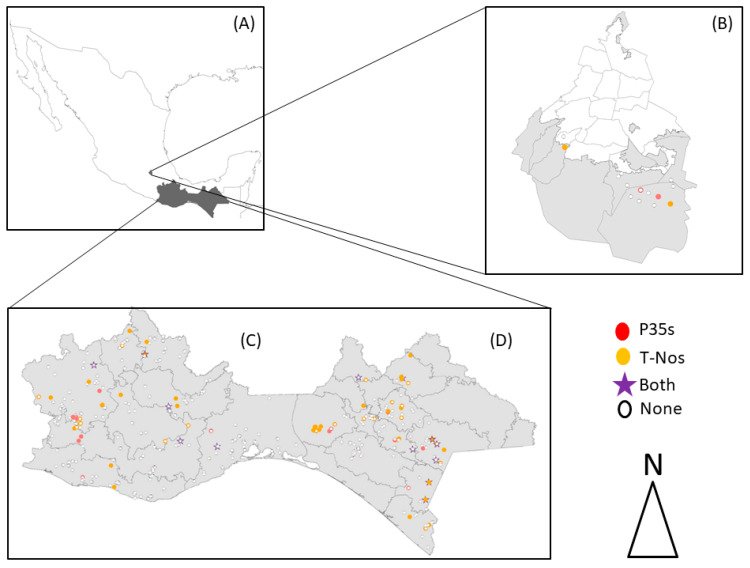
Graphic representation of transgene presence and type in the states of Mexico City, Oaxaca and Chiapas: (**A**) Map of Mexico with sampled states colorized in dark gray; (**B**) Localities sampled in Mexico City, with the limits between the 16 mayoralties that conform this state marked in black line. The gray area corresponds to the soil conservation area, where maize cultivation takes place in semi urban and rural communities; (**C**) Localities sampled in Oaxaca; (**D**) Localities sampled in Chiapas. For Oaxaca and Chiapas, we show their internal socioeconomic regions. Red circles represent localities where the 35S CaMV Promoter (P35S) was detected; yellow circles represent localities where the NOS terminator (T-NOS) was detected, purple stars represent localities where both transgenic markers were found and white circles represent localities with negative results to transgene presence. Localities with a combination of symbology mean that different results were found in nearby localities that due to the scale used in these maps, seem to overlap.

**Table 1 plants-12-02514-t001:** Agricultural practices and dynamics of surveyed maize producers. Comparison of seed management dynamics and cultural practices from producers of the three surveyed states. We grouped the responses according to the demographics of the maize producers and their production units, the characteristics of their maize seed lots and exchange dynamics and agricultural practices.

State	Mexico City	Chiapas	Oaxaca
** *Characteristics of maize producers and units of production* **			
Number of producers interviewed	115	391	400
Average age of producers	56.6	51.2	56.2
Percentage of producers over 50 years old	70.3	42	65
Percentage of producers who speak an Indigenous language	12.1	22.79	55
Type of land tenancy	communal, private	private, ejido	communal, private
Average extension (ha)	1.28	1.29	1.6
** *Characteristics of maize seed lots and exchange dynamics* **			
Average age of maize seed stock	more than 10 years	9.9	more than 10 years
% producers using only one variety of maize	36.52	82.90	75
% producers using more than one variety	63.47	17	25
% of samples ascribed as native maize varieties	98	77.6	91.1
% of samples ascribed as hybrid/improved varieties	2	22.4	8.9
% of seed exchanged within the locality	84.4	97.9	90.7
% seed introduced from outside of the locality	15.6	2.2	9.3
% of maize used for food	65.8	53.5	40.9
% of maize used for fodder	3.87	1.55	0.3
% of maize used for both	30.3	45	58.6
** *Agricultural practices* **			
% parcels sowed under monoculture	23.2	88.3	26.5
% parcels sowed under polyculture	76.8	11.7	73.5
% producers who use animal draft power	29.7	32.4	56.5
% producers who use a tractor	29.7	50	21.6
% producers who use both	40.7	17.6	33.5
% producers who apply fertilizers	53	100	62.8
% producers who apply herbicides	21	100	39.1
% producers who apply insecticides	7	67	23.5

**Table 2 plants-12-02514-t002:** Variables derived from our surveys that had a strong spatial association with transgene presence. *ε* > 2 represents significant spatial associations. Epsilon values equivalent or above 2 are bold, as well as values close to 2 are in bold. The variable “Origin of donated seed” refers to the origin of seed donated to the producer and used by her/him for cultivation. “*n*” refers to the number of producers that fell into that category.

State	Variable	Category	*n*	*ε*
Mexico City	Area (ha)	5	3	**2.58**
	Producer age group	(63, 67)	34	1.43
	Seed stock age	(25, 30)	21	1.30
	Producer age group	(54, 59)	22	1.21
Chiapas	Origin of donated seed	others	86	**2.00**
	Producer age group	(37.2, 43)	58	**1.87**
	Seed stock age	(5, 8)	89	**1.85**
	Language	native	113	1.35
	Seed stock age	(0, 2)	23	1.32
	Insecticide	others	241	1.12
	Herbicide (glyphosate)	other	3	1.07
Oaxaca	Producer age group	(43, 46)	33	**2.95**
	Use of harvested grain	sale	296	**2.37**
	Seed stock age	(5, 8)	36	**1.97**
	Producer age group	(54, 59)	56	**1.89**
	Origin of seed stock	seed shop	3	1.68
	Origin of donated seed	family	271	1.58
	Insecticide	derived from glyphosate	66	1.53
	Level of technification	tractor	361	1.45
	Cultivation system	polyculture	417	1.22
	Maize variety cultivated	hybrid	77	1.20
	Producer age group	(67, 74)	84	1.08

**Table 3 plants-12-02514-t003:** Transgene presence and distribution in different socioeconomic regions of Chiapas and Oaxaca. Average number of producers per region and percentage of positives for transgene presence, divided by producers, localities and regions in each of two states. There are five regions with the highest number of positive samples in Chiapas and three in Oaxaca (numbers in bold represent regions with the highest number of positive samples).

				Chiapas				
REGION	Number of Sampled Locations	Samples per Region	Number of Producers	Average Number of Producers per Region	Samples Positive for Transgenes	Percentage of Producers with Positive Samples	Percentage of Localities with Positive Samples	Percentage of Positive Samples within a Region
I. Metropolitana	20	76	64	6.4	**10**	2.56	50.00	13.16
II. Valles Zoque	10	34	28	2.8	**9**	2.3	90.00	26.47
III. Mezcalapa	3	3	3	0.3	0	0	0.00	0
IV. De los llanos	6	27	24	2.4	3	0.77	50.00	11.11
V. Altos Tsotsil-Tseltal	9	52	53	5.3	2	0.51	22.22	3.85
VI. La Fraylesca	16	44	38	3.8	3	0.77	18.75	6.82
VII. De los Bosques	2	14	12	1.2	3	0.77	66.66	21.43
VIII. Norte	4	20	18	1.8	1	0.26	25.00	5
IX. Istmo-Costa	3	10	9	0.9	0	0	0.00	0
X. Soconusco	17	42	32	3.2	**5**	1.28	29.41	11.9
XI. Sierra Mariscal	9	35	29	2.9	**7**	1.79	77.78	20
XII. Selva Lacandona	2	8	6	0.6	0	0	0.00	0
XIII. Maya	3	17	13	1.3	3	0.77	100.00	17.65
XIV. Tulijá Tsotsil Chol	6	19	14	1.4	2	0.51	33.33	10.53
XV. Meseta Comiteca	14	53	48	4.8	**8**	2.05	57.14	15.09
				**Oaxaca**				
**REGION**	**Number of Sampled Locations**	**Samples per Region**	**Number of Producers**	**Average Number of Producers per Region**	**Samples Positive for Transgenes**	**Percentage of Producers with Positive Samples**	**Percentage of Localities with Positive Samples**	**Percentage of Positive Samples within a Region**
I. Valles Centrales	8	63	41	3.28	2	0.5	25.00	3.17
II. Costa	15	74	58	4.64	2	0.5	13.33	2.7
III. Sierra Norte	11	42	23	1.84	**5**	1.25	45.45	11.9
IV. Sierra Sur	28	156	89	7.12	1	0.25	3.57	0.64
V. Cañada	12	45	15	1.2	2	0.5	16.67	4.44
VI. Mixteca	30	93	40	3.2	**12**	3	40.00	12.9
VII. Istmo	26	100	87	6.96	2	0.5	7.69	2
VIII. Papaloapan	12	66	47	3.76	**7**	1.75	58.33	10.61

## Data Availability

The anonymized producer data analyzed in this study are available on request from the corresponding author. The data are not publicly available due to privacy reasons.

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
