# Peer review of "Local and Regional Dynamics of Native Maize Seed Lot Use by Small-Scale Producers and Their Impact on Transgene Presence in Three Mexican States"

_plants, 2023, doi:10.3390/plants12132514_

Round 1

Reviewer 1 Report

Ayala-Angulo et al. have well-written this article. The authors have discussed the results of their surveys and tried to hypothesize how detectable levels of transgenes could have been introduced to small-scale farming across South Mexico. The authors have identified and hypothesized the relationship between the introduction of transgenes with demographic and age distribution of farmers, seed exchange rate between them, seed management behavior, and other factors. These data could have been published along with Rendon-Aguilar et al. (2019) article. It is commendable that the authors have decided to make it a separate report/article. Although the introduction and discussion parts are detailed, it felt like some of the discussed paragraphs were not to the point and too long to comprehend. I request the authors to summarize most of the sections to make reading easier. If possible, please remove unrelated sentences. 
It is a well-known fact that seed movement through human activities can cause the introduction of unwanted genotypes in a farming region. However, using unpublished data and personal opinions should be avoided to make a point. Please review the paragraph (lines 399-411). 

One crucial thing that can be noticed in this article is the use of the font. The fonts used in this article are not similar, especially after line 326. Please review. 

A few grammatical errors should be improved. 

Author Response

We appreciate the observations and suggestions made by reviewer 1. Please see the attachment for an itemized response.

Reviewer 2 Report

This is a good study. The paper investigated the presence of transgenes in the native maize varieties in Mexico, regarding the impacts of characteristics of farmers and farming practices. The authors suggested good explanations of their results and provided suitable strategies to reduce the presence of GM maize in local production. I would suggest publishing it with priority in the journal.

It is interesting to see that farmer’s age is correlated to the presence of GM contamination. I understand that the younger farmers may like new maize cultivars, while the older farmers may prefer the cultivars of high production. Use of machinery may also be associated with the presence of transgene. Can the authors further elucidate on these issues in the main text?

Seed exchange may result in transgene flow, is there possibility of transgene escape by pollens? As the paper implicated, the native varieties are of white color while the foreign ones from USA are normally in yellow. Did farmers care about the grain color? The attitude may affect the percentage of transgene presence. Will be useful to educate farmers to take care to reduce the possibility of transgene flow?

Author Response

We appreciate the observations and suggestions made by reviewer 2. Please see the attachment for an itemized response. 
